# Preparation and Mechanical Properties of Flexible Prepreg Resin with High Strength and Low Creep

**DOI:** 10.3390/polym16040558

**Published:** 2024-02-18

**Authors:** Zhaoyi Sun, Zhiyuan Mei, Zheng Huang, Guorong Wang

**Affiliations:** Faculty of Warships and Oceanography, Naval University of Engineering, Wuhan 430033, China; Zhaoyi_sun0805@163.com (Z.S.); huangzheng.315@163.com (Z.H.); guorong_wang1995@163.com (G.W.)

**Keywords:** medium–low modulus, prepreg resin, high strength, bending creep

## Abstract

In this study, aiming at the problem of low strength and high creep caused by medium–low modulus flexible resin based on the formulation design idea of high-molecular-weight epoxy resin (E12)-reinforced flexible epoxy-terminated urethane resin (EUR), a flexible epoxy prepreg resin with high strength and low bending creep was prepared to be suitable for hot melt processing technology. Flexible EUR was synthesized by grafting flexible polyurethane segments onto the epoxy side chain by urethane bonding. By adjusting the ratio of E12 and EUR, the effects of different ratios of the two components on the mechanical properties and viscoelasticity of the resin were systematically studied with dicyandiamide as the latent curing system. Research has found that when the E12 content is between 20%wt and 40%wt, the resin system has the best coating viscosity at 65 °C to 85 °C. The molecular weight and the content of aromatic heterocyclic groups of the resin determine the strength and creep behavior of the resin. When the content of E12 in the system is less than 50%wt, modulus and strength increase linearly, but after more than 50%wt E12 content, the modulus is almost unchanged and the strength begins to decrease. By increasing the content of E12 in the resin, the creep behavior of the resin is greatly reduced. When the content of E12 increases to 50%wt, the bending creep is the lowest.

## 1. Introduction

Fiber-reinforced polymer matrix composites have the advantages of high specific strength, good damping, seawater corrosion resistance, and non-magnetic characteristics, which has resulted in many researchers attempting to apply the composites to marine propellers [1,2,3,4]. By realizing the large deformation swing of the composite blade, the unsteady force is reduced, the excitation force is reduced, and the radiated noise of the propeller is reduced [5,6]. In order to realize the large deformation swing of the blade, the blade needs to be designed with low stiffness, which puts forward new performance requirements for traditional composite laminated materials, such as medium and low modulus, low creep, and high strength. However, the anisotropy of the composite material is a major feature of its free design [7]. Medium and low modulus composites provide enough optimization design space for structural designers to realize low-stiffness propeller design. Through the rational use of transverse modulus and strength parameters, structural designers optimize the design of each layer angle and can control the strength and stiffness of the propeller guide and trailing edges in different directions [8,9]. Thus, it can fully ensure the strength of the safety structure and meet the special functional requirements of the structure. Nowadays, the composite materials of high-quality products are all prepared by prepreg, which has many advantages, such as simple molding and strong reliability, and is suitable for the manufacture of complex specially shaped structures [10,11,12]. However, the current commercially available aviation-grade prepregs often pursue high-temperature resistance and high modulus performance, which makes it difficult for designers to perform structural ply design for the large pre-deformation effect at the blade tip. Therefore, it is necessary to carry out research on flexible prepreg resin.

According to the classical laminate theory (CLT) [13,14,15], the modulus in the transverse direction is mainly determined by the modulus of the resin matrix and the fiber volume fraction. Among them, the fiber volume fraction is achieved by adjusting the thickness of the resin film and the fiber yarn bundle, and the resin modulus is controlled by the molecular structure design of different soft and hard segments and crosslinking density in the material formulation. In the processing of prepregs, the thickness of the resin film is too high to overflow, and too little fiber content will lead to open cracks in prepregs. Therefore, the range of transverse modulus of composite materials adjusted by processing technology is limited. In engineering applications, most of the transverse tensile properties are reasonably controlled by the selection of resin materials. At present, the prepreg prepared by the hot melt film method [16,17] has the advantages of less volatile matter, accurate control of resin content, good product quality and balance, and it is the most widely used preparation method. The hot melt film method has strict requirements for prepreg resin, which needs to meet the requirements of appropriate paving viscosity at room temperature, appropriate film viscosity after heating, and sufficient storage and shelf life at room temperature [18,19]. Epoxy resin has been widely used by researchers because of its good strength, adhesion, and water resistance [20,21,22]. It is often difficult for a single epoxy resin to meet the preparation and processing technology of prepregs, and epoxy resins with different properties are needed to meet the multi-performance requirements of composites. Researchers are mostly committed to solving the problem of the high brittleness of epoxy resin. There are many modification methods for toughening epoxy resin, such as adding a modified toughening agent [23,24], core–shell rubber particle modification [25], thermoplastic modification [26], hyperbranched polymer modification [27,28], adding inorganic/organic nanoparticles [29,30], flexible chain polymers [31], and other means to toughen epoxy. At present, the toughening methods of epoxy resin have been studied in depth, but few researchers pay attention to the preparation of flexible epoxy prepreg resin, as well as the low strength and large creep caused by the decrease in modulus.

In this study, polyurethane flexible segments were introduced into the side chain of brittle epoxy resin and blended with high molecular weight solid epoxy resin. By increasing the molecular weight of the resin system and the content of aromatic heterocyclic groups, the strength and low creep of the flexible resin were improved, and the processing technology of hot melt prepreg resin was satisfied. Firstly, the NCO-terminated polyurethane prepolymer was obtained by the addition reaction of bifunctional polyether and toluene diisocyanate, and the epoxy-terminated polyurethane flexible resin was synthesized by the urethane reaction with the hydroxyl group on the epoxy side chain. After that, the resin was used as the main resin to compound with high-molecular-weight epoxy resin, and the latent curing agent dicyandiamide and the latent accelerator imidazole derivative were used to cure the resin. The addition of high-molecular-weight epoxy greatly increases the viscosity of the resin system. At the same time, the increase in molecular weight and aromatic heterocyclic group content of the resin has a certain improvement in modulus, strength, and anti-creep.

## 2. Experimental Methods

### 2.1. Materials

Polypropylene oxide glycol (molecular weight 2000) was purchased from Shanghai McLin Biochemical Technology Co., Ltd. (Shanghai, China). Toluene diisocyanate (TDI-80) was purchased from Shandong Bai qian Chemical Co., Ltd. (Jinan, China). Epoxy resin (E44, epoxy equivalent (EE) = 227 g/mol) was purchased from Yue yang Petrochemical Complex Yuehua Organic Chemical Plant (Yueyang, China). High molecular weight bisphenol A epoxy resin (E12, epoxy equivalent (EE) = 833 g/mol) was purchased from Baling Petrochemical Co., Ltd. (Yueyang, China). Latent curing agent dicyandiamide (DICY, solid powder particle size of 5–10 microns) was purchased from Shanghai Aladdin Biochemical Technology Co., Ltd. (Shanghai, China). The latent curing accelerator is a commercially available imidazole derivative (GYHT110, solid powder particle size of 2~5 microns) purchased from Guangdong Guyan Electronic Materials Co., Ltd. (Qingyuan, China). Defoamer (Deform6800) was purchased from Dow Corning (Midland, MI, USA).

### 2.2. Synthesis of Epoxy-Terminated Urethane Resin (EUR)

The polyurethane flexible segment was grafted onto the epoxy resin main chain segment by molecular design. According to the calculated amount (NCO/OH molar ratio 2.1:1), the required amount of PPG2000 was weighed and poured into a three-port bottle containing a stirrer and a thermometer. Under the protection of nitrogen, vacuum distillation was dehydrated for 2 h. After that, the PPG2000 material was stirred after the temperature was controlled at 80 °C, and the calculated amount of TDI-80 was added to the constant pressure drip funnel and slowly added to the three-port bottle. The reaction temperature in the bottle was controlled between 80 °C and 90 °C, and the reaction temperature was gradually increased to 105~110 °C to continue the reaction. In the reaction process, after the system reacted for 1 h, the sample was taken and the content of -NCO in the system was continuously determined by chemical analysis until the theoretical value was reached (the theoretical value was calculated according to the amount of PPG and TDI). The polyurethane prepolymer (PU prepolymer) was synthesized as the intermediate of EUR. When the PU prepolymer was cooled to 80~95 °C, E44 with the calculated amount (molar value of 1.1 mol) was added to the prepolymer. Then, the reaction temperature of the system was slowly raised to 105~110 °C to continue the reaction, continue high-speed stirring, and continue to measure the -NCO content in the system until the -NCO content in the system reached the theoretical value of 0% (generally the measured value is 0.05~0.1%), indicating that the synthesis reaction is over. The epoxy equivalent of EUR was determined by GB/T 4612-2008 standard [32], EE = 350.87 g/mol. The specific synthesis reaction mechanism of the EUR is shown in Figure 1.

### 2.3. Preparation of Prepreg Resin

The synthesized EUR and high-molecular-weight epoxy resin (E12) were placed in a beaker according to the formula ratio of Table 1, melted at 110 °C, and mechanically stirred at 300 rpm/min for 2 h until the two resins were completely mixed. After the blending resin was cooled to 75 °C, DICY, GYHT110, and Deform6800 were added, and the dispersion plate was stirred at a speed of 1000 rpm/min for 20 min. The resin was placed in a vacuum drying oven at 80 °C, defoamed for 15 min, and poured into a preheated steel mold. After pouring, it was placed in an oven and cured according to the temperature rise curve in Figure 2.

### 2.4. Characterization and Performance Testing

A Fourier-transform infrared spectrometer (TENSOR II, Bruker (Beijing) Technology Co., Ltd., Beijing, China) was used to characterize the groups in each stage of resin preparation. Test conditions: the test resolution is 4 cm^−1^, the number of scans is 16 times, and the test range is 400~4000 cm^−1^. IR spectra were preprocessed using the infrared analysis software OMNIC 8.2, including baseline correction, 5-point smoothing, and normalization (minimum/maximum normalization).

The viscosity–temperature characteristics of prepreg resin were tested by a BROOKFIELD CAP 2000+ high shear cone plate viscometer. The test temperature range was 60~90 °C, and the viscosity was tested once every 5 °C. Each group of samples was tested three times.

A universal tensile testing machine (CMT4204, Meters Industrial System (China) Co., Ltd., Shanghai, China) was used to test the tensile strength of the sample according to the GB/T 528-2009 standard [33]. The loading rate was 2 mm/min, and the mean value of each sample was taken 5 times. According to the GB/T 9341-2008 standard [34], the bending strength of the sample was tested. The loading rate was 2 mm/min, and the sample size was 80 mm × 10 mm × 4 mm. Each sample group was tested 5 times and the mean value was taken.

The plastic pendulum impact tester (PTM1000) was used for testing. The impact test of plastic cantilever beams was carried out according to GB/T 1843-2008 standard for the determination of unnotched impact strength [35]. The sample size was 80 mm × 10 mm × 4 mm, and the mean value of each sample group was tested 5 times.

Microscopic analysis of ductile fracture: The microscopic interface of the fracture notch of the impact specimen was observed on a Sigma300 scanning electron microscope (SEM, Carl Zeiss AG, Oberkochen, Germany). Before observation, the sample was gilded.

The molecular weight of the resin was analyzed by an Ailment 1100 high-performance gel permeation liquid chromatograph (Agilent, California, CA, USA) and RID differential detector. According to the molecular weight, the standard styrene solution with a mass fraction of 0.05~0.5% was prepared as the standard sample, and chloroform was selected as the mobile phase. The flow rate of the mobile phase was selected as follows: under the condition of column temperature 40 °C and injection volume 20 μL, the flow rate was inversely proportional to the number of trays. In order to take into account the determination time and column efficiency, the flow rate was selected as 0.35 mL/min.

The dynamic mechanical temperature spectrum of the sample was measured using a dynamic thermomechanical analyzer (DMA1, METTLER TOLEDO, Zurich, Switzerland). The test mode is single cantilever mode, the test frequency is 1 Hz, the heating rate is 3 °C/min, and the temperature scanning range is 0~180 °C.

The bending creep properties of the samples were measured using a dynamic thermomechanical analyzer (DMA1, METTLER TOLEDO, Zurich, Switzerland). The test mode is three-point bending mode, the test temperature is 35 °C, the stress load is 1.0 MPa, and the scanning time is 21,000 s.

## 3. Results and Discussion

### 3.1. FTIR and Molecular Weight Analysis of the Resin

The group reaction during the synthesis of epoxy-terminated urethane resin was observed by FTIR, as shown in Figure 3a. It can be seen from the figure that the -NCO group of the polyurethane prepolymer has a strong stretching vibration absorption peak at 2250 cm^−1^, while the synthetic product EUR has no absorption peak at this wave number, which indicates that the -NCO group has completely reacted with the -OH group on the main chain of the epoxy resin. The -OH is completely reacted to obtain the urethane bond so that the polyurethane side chain is successfully grafted onto the epoxy resin main chain (the synthesis reaction mechanism is shown in Figure 1). The absorption peak of the stretching vibration of -OH at the wave number 3650~3200 cm^−1^ in the infrared spectrum of the EUR is weaker than that of epoxy resin, and the reaction between groups can also be seen. From Figure 3b, we found that with the increase in E12 content, the peak of 3080~3010 cm^−1^ in the spectrum gradually increased. This is because the content of the aromatic hydrogen group (-CH) in the aromatic heterocyclic skeleton on the benzene ring increased so that the stretching vibration peak was enhanced at this wavenumber. At the same time, the C-C on the aromatic ring skeleton has 2~4 peaks at 1610~1450 cm^−1^, and the stretching vibration peak at 1610~1450 cm^−1^ was also found to be enhanced from the spectrum. The stretching vibration peaks of the two wavenumber segments are enhanced, which proves that the increase in E12 content increases the content of the overall aromatic heterocyclic group of the resin. It is worth noting that since the epoxy equivalent of the synthesized EUR resin (EE = 350.87 g/mol) is lower than that of E12, the epoxy group content of the resin should decrease with the increase in E12 content, which is demonstrated by the weakening of the stretching vibration peak at 916 cm^−1^ in the spectrum.

The molecular weight of the polymer often determines the processing viscosity and mechanical properties of the material. We tested the molecular weight of different formulations of the resin by gel permeation chromatography, and the test results are listed in Table 2. From the data in the table, it can be seen that the molecular weight of the resin increases linearly with the increase in E12 content, which is consistent with our conjecture. The data from the table also show that the resin is a polydispersity sample because the polydispersity coefficient (Mw/Mn) of the resin is not equal to 1.0. The larger the polydispersity coefficient, the wider the molecular weight dispersion. However, the degree of dispersion of the molecular weight of the resin does not increase regularly with the increase in E12 content, which may be due to multi-component polymers with different molecular weights in the resin. Generally speaking, the size of the molecular weight often determines the viscosity and strength of the resin. The higher the molecular weight of the resin, the greater the viscosity and the higher the strength, which is reflected in the analysis of viscosity and mechanical properties in Figure 4 and Figure 5.

### 3.2. Process Analysis of Prepreg Resin

The viscosity–temperature characteristics of resin are an important index parameter in the production of prepreg processing. When the resin is hot-melted through the production equipment, it is difficult to form a continuous film on the release paper due to the high viscosity of the resin, and the low viscosity of the resin causes it to easily overflow. Therefore, the resin has moderate viscosity parameters, which is very important for the production of qualified prepregs. In this paper, the viscosity of different formulations of resin was tested in the temperature range of 60~90 °C by using a high shear cone plate rotary viscometer. The test results are shown in Figure 4. Generally speaking, the viscosity of 18,000~40,000 cPs can be coated, and the best coating viscosity is 20,000 cPs. When the fiber and film are penetrated, the resin needs to have good fluidity, so the viscosity should be below 10,000 cPs. Figure 4 shows that as the amount of epoxy E12 increases, the viscosity of the resin system increases to varying degrees at different temperatures. In the temperature range of 65~85 °C, when the amount of E12 is 20~40%wt, the viscosity of the system is between 10,000~40,000 cPs, and the viscosity change is the largest, which provides a favorable choice for the coating process parameters. It is worth noting that when the resin system is greater than 85 °C, the viscosity of the resin system with an E12 dosage of 20~40%wt is less than 10,000 cPs, and the resin has good fluidity and can infiltrate the fiber well.

### 3.3. Tensile, Bending, and Impact Resistance Analysis

Figure 5a shows the tensile stress–strain curve of the resin. It can be seen from the curve change trend that almost all the resins in the formula break in the tensile plastic deformation area, showing different degrees of ductile fracture. Specific data results are shown in Table 3. The results show that the synthesized EUR has the lowest Young’s modulus and tensile strength, which are 1.227 GPa and 32.68 MPa, respectively. With the increase in E12 content, the tensile strength and Young’s modulus of the resin first increased and then decreased. When the content of E12 was 20%wt, the Young’s modulus and tensile strength increased to 1.748 GPa and 44.7 MPa, respectively. As its amount continues to increase to 50%wt, the Young’s modulus and tensile strength increase to a maximum, which are 2.416 GPa and 57.6 MPa, respectively. This is mainly due to the high content of aromatic heterocycles in the E12 molecular chain. At the same time, from the molecular weight test results of Table 2, it was found that the molecular weight of the resin gradually increases, so the strength and modulus increase. It is worth noting that when the amount of high-molecular-weight epoxy is increased, the Young’s modulus of the system hardly increases. On the contrary, the tensile strength decreases. This may be because the ratio of EUR to E12 is interchangeable, and the density of polar groups is too large, so the molecular chain exhibits brittle fractures, resulting in a decrease in strength. The tensile strength and ductility of the flexible prepreg resin in this study are higher than those of toughened epoxy resins reported in many studies [30,36,37,38].

The bending performance of six groups of specimens in the formula was tested. The bending stress–strain curve is shown in Figure 5b. Specific bending performance data are shown in Table 3. From the comparison of the data, with the increase in the amount of high-molecular-weight epoxy, the bending strength and bending modulus of the system are improved, and this trend is consistent with the tensile properties. The bending strength of EUR is only 37.9 MPa. When 20%wt of E12 is added, the strength of the system increases to 59.8 MPa, and the strength increases by 58%. When its dosage reaches 50%wt, the bending strength reaches a maximum value of 91.3 MPa, which is 141% higher than that of EUR. This is mainly due to the increase in E12 content, which increases the rigid aromatic heterocyclic groups contained in the main chain, and also increases the polarity of the molecular chain. The increase in the density of polar groups can increase strength. In addition, when the amount of E12 exceeds 50%wt, the bending strength and tensile strength change consistently and begin to decrease.

Figure 6 shows the impact strength of resin systems with different formulations. There are many types of impact strength tests. In this study, the pendulum impact test was used to test the toughness of samples with different formulations. The toughness strength is based on the material before fracture and its absorption of energy; it is the full embodiment of material strength and ductility. From the tensile stress–strain curve in Figure 5a, it can be seen that although the elongation at break of 0%-E12 is the highest, the impact strength is not the best, and the strength is only 25.1 kJ·m^−2^. When the content of E12 in the system is 20%wt, the overall molecular weight of the system increases, the toughness increases, the impact strength is as high as 35.8 kJ·m^−2^, and the toughness increases by 42.6%. Generally, the molecular weight of the polymer increases, and the impact strength tends to increase. However, when the amount of E12 exceeds 20%wt, the toughness strength decreases rapidly. This may be due to the excessive content of aromatic heterocycles in the main chain, which reduces the ductility of the resin, makes the molecular chain less prone to relative slip, increases brittleness, and thus reduces toughness, which can be reflected in the morphology of the impact fracture surface. When the amount of high-molecular-weight epoxy exceeds 50%wt, the toughness of the resin is the worst, and the impact strength is only 5.1 kJ·m^−2^.

### 3.4. Microscopic Analysis of Impact Fracture Surface

The impact fracture surface was analyzed by scanning electron microscopy, and a further toughness mechanism was obtained. The fracture morphology is shown in Figure 7. Figure 7f shows a relatively smooth fracture surface, highlighting a brittle fracture surface, which also reflects the poor toughness strength of 60%-E12 in Figure 6. In contrast, Figure 7a–d show a rough fracture morphology, in which Figure 7b shows more cavities, with the highest fracture roughness. And the molecules absorb a lot of energy in the elastic zone and plastic deformation zone after being subjected to impact load, which corresponds to the best toughness strength of 20%-E12. Due to the high molecular weight of epoxy containing 20%wt, a sea–island structure is formed, showing “plastic–rubber” phase separation, forming a large number of holes, and hindering the expansion of cracks. In addition, SEM micrographs of the fracture surface show the appearance of deformation cavitation related to crack passivation and termination. The amount of cracks and ductile fibrillation can absorb fracture energy, thus improving the impact resistance of cured resin.

### 3.5. Viscoelastic Analysis

The dynamic viscoelasticity of the resin was evaluated by a dynamic thermodynamic performance test. Generally speaking, viscoelasticity is an important characteristic of polymer materials. Changes in mechanical properties over time are collectively referred to as mechanical relaxation. According to the different external loads on the material, different mechanical relaxation phenomena can be observed, mainly including creep, stress relaxation, hysteresis, and mechanical loss. Figure 8 shows the change curves for storage modulus, loss modulus, and loss factor of different resin formulations. From the curve in Figure 8c, it is shown that the glass transition temperature (T_g_) of the sample 0%-E12 is the lowest, only 93.07 °C. With the increase in E12 content, the T_g_ of the system increases. This may be due to the addition of high molecular weight resin, which increases the molecular weight of the entire molecular chain of the system and improves the storage modulus of the system, which is reflected in the storage modulus spectrum of Figure 8a. It is worth noting that the loss modulus of the resin system increases significantly from the change curve of the loss modulus in Figure 8b. The sample loss modulus peak value of 0%-E12 is the lowest, and with the increase in the additional amount of E12, the loss modulus peak value increases, and the peak temperature also increases. This may be due to the increase in the chain length of the molecular chain of the system so that the molecular chain is subjected to the action of internal friction when it moves in the viscoelastic region at high temperatures. Under the alternating stress load, the molecular chain movement cannot keep up with the change in the external force load, and the deformation lags behind the stress with a phase difference, and the hysteresis phenomenon occurs. In each cyclical change, merit is consumed and mechanical loss is generated. Due to the addition of high-molecular-weight epoxy resin, the length of the molecular chain increases, and curl between the chains occurs, resulting in higher mechanical loss in the viscoelastic region.

Using the three-point bending mode of the dynamic mechanical analyzer, the short-term bending creep characteristics of the sample at 35 °C were tested. The test results are shown in Figure 9. From the curves in the figure, it can be found that the three stages of creep (instantaneous deformation, primary creep, and secondary creep) are obvious. There is no third-order creep in the curve. Third-order creep occurs only at a high stress value or at this stress level for a long time [39,40]. As expected, the increase in high-molecular-weight epoxy content weakens the bending creep behavior of the resin. When the content of E12 is 0%wt, the bending strain increment is the largest, and the strain increment is 148.47%. When the content of E12 is 20%wt, the increase in bending strain decreases rapidly, and the increase in strain is 81.03%. With the increase in E12 content, the bending strain increment continues to decrease. When the E12 content is 50%wt, the strain increment is minimal, and the strain increase is only 29.29%.

### 3.6. Analysis of Creep Reduction Mechanism

From the results of the molecular weight test of the resin by gel permeation chromatography (GPC) in Table 2, it can be seen that the addition of high-molecular-weight epoxy makes the molecular weight of the resin system gradually increase, and the degree of the curl of the molecular chain increases. At the same time, as shown in Figure 3b, the stretching vibration peaks at 3080~3010 cm^−1^ (-CH on the aromatic heterocyclic skeleton) and 1610~1450 cm^−1^ (C-C on the aromatic heterocyclic skeleton) gradually increased. We determined that the content of aromatic heterocyclic groups in the resin molecular chain increased. By increasing the molecular weight and the content of aromatic heterocyclic groups, the steric blocking effect of aromatic heterocyclic groups in the molecular chain and the interlocking effect between curled molecular chain segments make it difficult for molecular chains to slip relative to each other, thereby improving the creep resistance of the resin under small loads.

In addition, we also found that the resin’s high damping loss and low creep properties are contradictory. In other words, resin reduces creep and also reduces damping loss. It can be seen from the curve in Figure 8b that the resin loss modulus in the glassy state determines the size of the bending creep behavior of the resin. This is due to the increase in molecular weight, which makes the molecular chain curl more. After being subjected to external force, the macromolecular groups of the chain and the chain hinder the movement of the molecular chain segment. With the increase in time, the deformation is small, the molecular chain segment shows general elastic behavior in the short term, the energy loss between the molecular chains is reduced, and the permanent deformation is weakened, showing good creep resistance.

## 4. Conclusions

This study presents a synthetic design idea of a high-strength, low-creep flexible epoxy prepreg resin that conforms to the hot melt process. Flexible epoxy-terminated urethane resin was synthesized by grafting the polyurethane soft segment to the epoxy main segment by urethane bond. The viscosity and strength of the prepreg resin were adjusted by using high molecular weight solid epoxy as a thickener and reinforcing agent. At the same time, the creep of the flexible prepreg resin was improved by using aromatic heterocyclic groups in the main chain of E12 and by increasing the molecular weight of the resin. With the increase in E12 content, the viscosity of the resin increases linearly. When the E12 content is 20~40%wt, there is an optimal viscosity of the hot melt coating. The strength and modulus of the resin increase with the increase in E12 content. When the E12 content is 50%wt, the strength reaches the maximum value. After the content continues to increase, the strength begins to decrease, and the modulus does not change significantly. Furthermore, E12 content is negatively correlated with creep strain increment. When the E12 content is 50%wt, the creep strain increment is minimal. Although from the point of view of mechanical properties, 50%wt-E12 has the highest strength and lowest creep. However, in order to prepare prepreg tape with excellent quality, it is necessary to focus on processing technology. The 40%wt-E12 has the best coating viscosity; its strength and low creep properties are second only to 50%wt-E12, and it has a lower modulus. Considering the processing technology and mechanical properties of the resin, 40%wt-E12 is the optimal formulation for the preparation of flexible prepreg resin. The preparation formula of medium and low modulus flexible prepreg resin developed in this paper provides a favorable material choice for the manufacture of low-noise composite propeller blades.

## Figures and Tables

**Figure 1 polymers-16-00558-f001:**
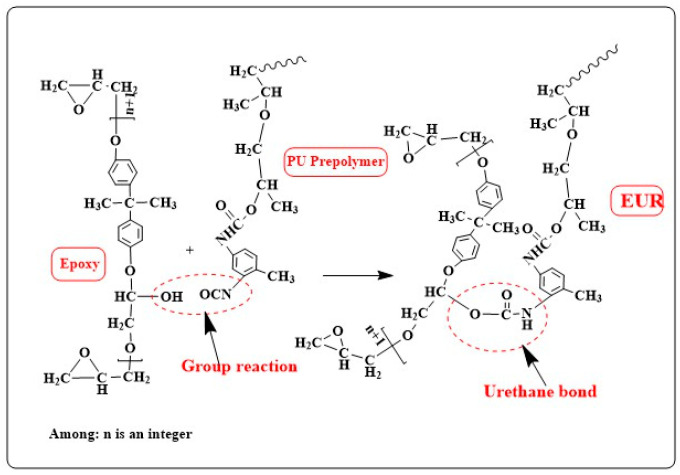
The EUR synthesis reaction mechanism diagram.

**Figure 2 polymers-16-00558-f002:**
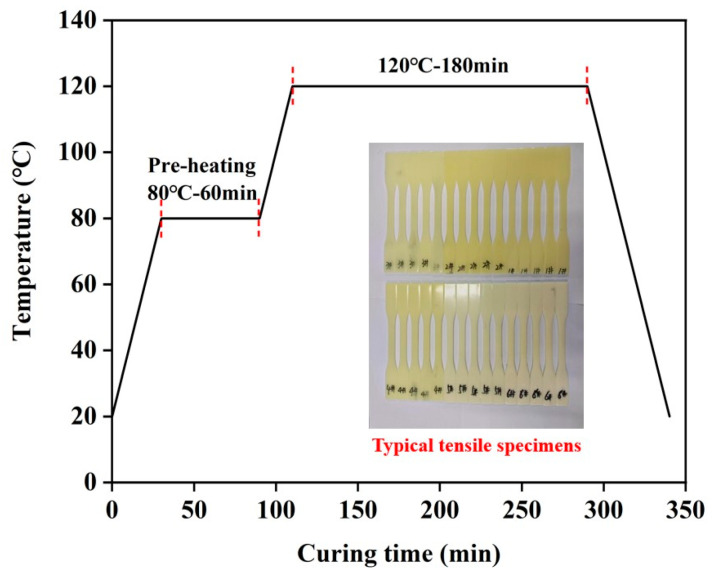
The curing process of prepreg resin.

**Figure 3 polymers-16-00558-f003:**
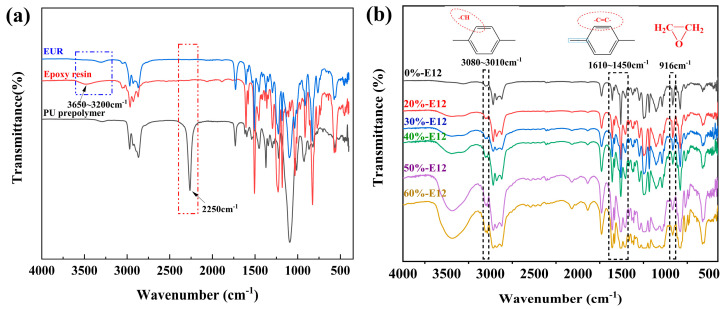
FTIR spectra: (**a**) EUR synthesis reaction; (**b**) prepreg resins with different E12 content.

**Figure 4 polymers-16-00558-f004:**
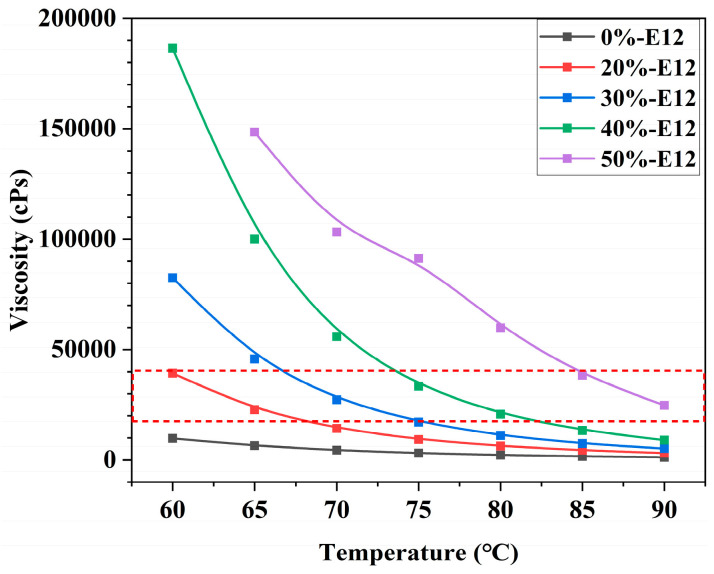
Viscosity vs. temperature curve of resin.

**Figure 5 polymers-16-00558-f005:**
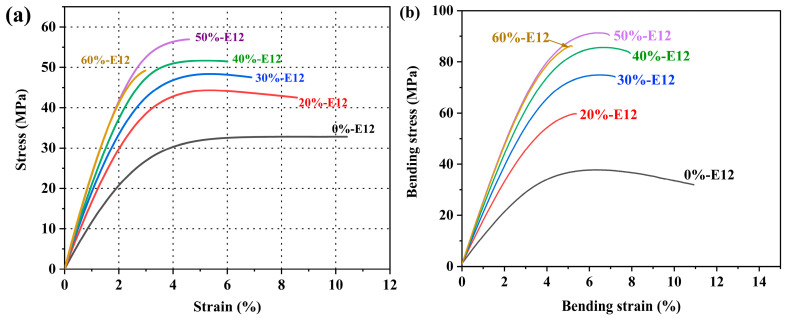
Tensile stress–strain curve (**a**) and bending stress–strain curve (**b**) of different resin formulations.

**Figure 6 polymers-16-00558-f006:**
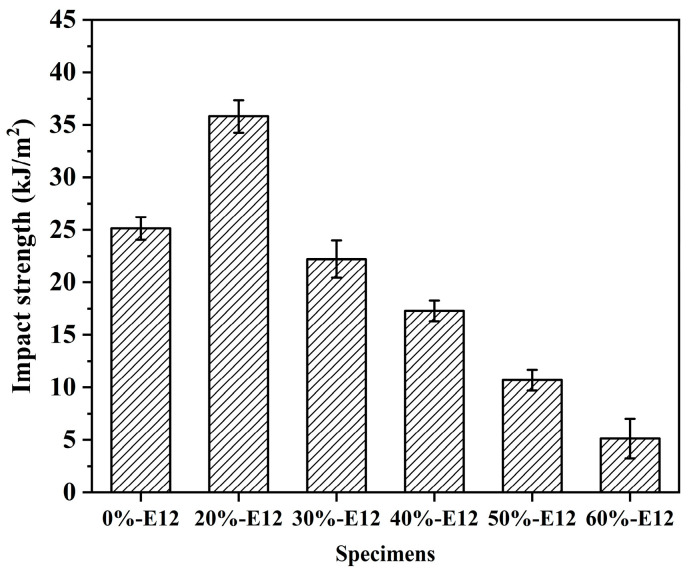
Impact strength of different formulations of resin.

**Figure 7 polymers-16-00558-f007:**
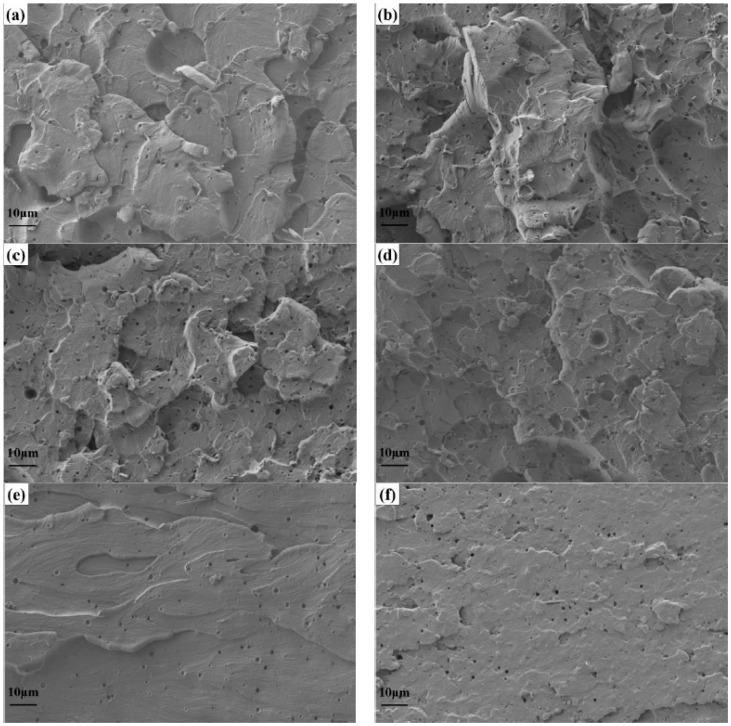
SEM images of impact fracture morphology: (**a**–**f**) is 0%-E12~60%-E12 in turn.

**Figure 8 polymers-16-00558-f008:**
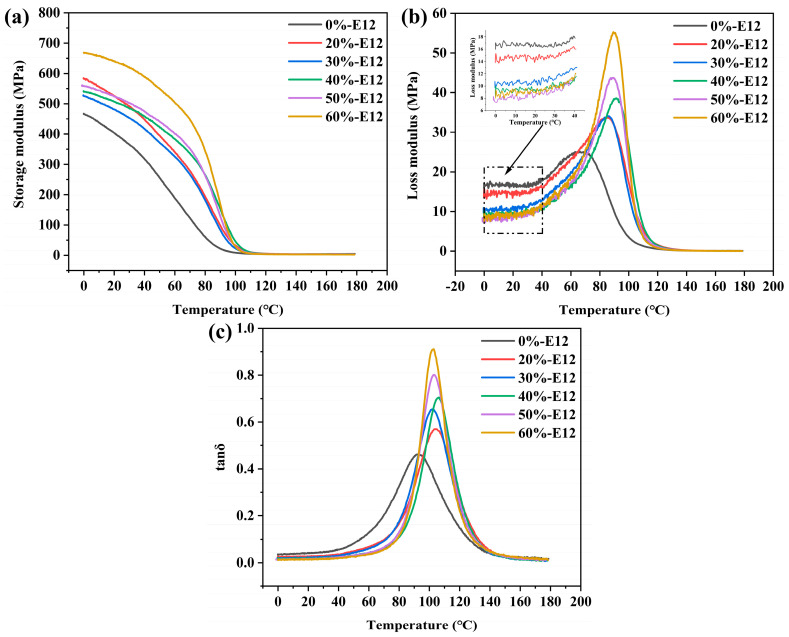
Changes in storage modulus curves (**a**); loss modulus curves (**b**); and loss factor curves (**c**) for different resin formulations.

**Figure 9 polymers-16-00558-f009:**
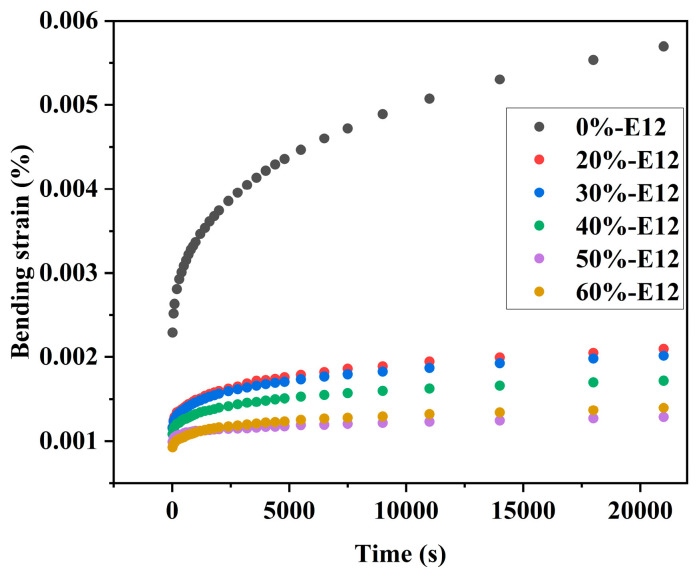
Effect of E12 content on bending creep characteristics.

**Table 1 polymers-16-00558-t001:** The formulation of prepreg resin.

Specimen	0%-E12	20%-E12	30%-E12	40%-E12	50%-E12	60%-E12
EUR	100	80	70	60	50	40
E12	0	20	30	40	50	60
DICY	4	4	4	4	4	4
GYHT110	1	1	1	1	1	1
Deform6800	0.3	0.3	0.3	0.3	0.3	0.3

**Table 2 polymers-16-00558-t002:** Molecular weight test results of different resin formulations (GPC test results).

Specimen	Mn	Mw	Mz	Mw/Mn
0%-E12	1294	1372	1479	1.06
20%-E12	3256	5714	8528	1.75
30%-E12	4207	6224	9575	1.48
40%-E12	5326	7046	10,172	1.32
50%-E12	6587	8407	11,952	1.28
60%-E12	7199	9001	12,874	1.25

**Table 3 polymers-16-00558-t003:** Mechanical property data of EUR and E12 resins with different formulations.

Specimen	0%-E12	20%-E12	30%-E12	40%-E12	50%-E12	60%-E12
Tensile strength/MPa	32.7 ± 0.4	44.7 ± 0.3	48.2 ± 0.5	52.1 ± 0.4	57.6 ± 0.4	49.3 ± 1.9
Young’s modulus/MPa	1227 ± 19	1748 ± 14	2001 ± 78	2181 ± 46	2416 ± 66	2420 ± 121
Bending strength/MPa	37.9 ± 0.7	59.8 ± 1.1	74.9 ± 0.8	85.7 ± 0.7	91.3 ± 1.2	86.3 ± 2.1
Bending modulus/MPa	1124 ± 35	1759 ± 27	2073 ± 53	2307 ± 22	2498 ± 35	2445 ± 49
Impact strength/kJ·m^−2^	25.1 ± 1.1	35.8 ± 1.6	22.2 ± 1.8	17.3 ± 1.0	10.7 ± 1.0	5.1 ± 1.8

## Data Availability

The data presented in this study are available in the article.

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
