# Peer review of "Preparation and Mechanical Properties of Flexible Prepreg Resin with High Strength and Low Creep"

_polymers, 2024, doi:10.3390/polym16040558_

Round 1

Reviewer 1 Report

Comments and Suggestions for Authors

REVIEW of the article "Preparation and mechanical properties of flexible prepreg resin with high strength and low creep" by Zhaoyi Sun, Zhiyuan Mei, Zheng Huang and Guorong Wang

At present, the study of the possibility of using polymeric materials to obtain products operating under high loads and often in aggressive environments is an urgent problem. In the presented article the authors propose to carry out modification of epoxy resin in order to increase mechanical characteristics and reduce creep. The paper presents a rather extensive literature analysis on the application of the developed materials for marine propellers, and also presents modern and complementary methods of composites research.

The following comments are made on the article:

1. The authors measured the viscosity of the developed material to determine the optimal composition with good flowability and ability to penetrate the fiber. However, there is no mention anywhere: which fiber is planned to be used for impregnation and what are the adhesion characteristics between the fiber and the developed material?

2. What explains the significant increase in viscosity at 40%wt-E12
compared to 30%wt-E12 and 50%wt-E12 (Figure 4)?

3. The authors' statement that "In the temperature range of 65°C-85°C, when the amount of E12 is 20%wt-40%wt, the viscosity of the system is between 10000cPs-40000cPs" (lines 211-213) is true only for 40%wt-E12. For 20%wt-E12 and 30%wt-E12, the viscosity does not reach these values at any temperature.

4. Very little attention is paid to Table 3 in the article, although it is the change in molecular weight that largely determines the change in mechanical properties. It is reasonable, in my opinion, to move Table 3 after the FT-IR data and discuss it in more detail.

5. Is the developed material planned to be used for marine propeller blades? If yes, why such important parameters as density of the developed material, its water absorption and resistance to corrosion in sea water are not considered?

6. Line 221: there is a typo, figure 5(a) should be indicated.

7. It is recommended to expand the conclusions, as it is not clear what formulation of the developed material the authors ultimately recommend.

Reviewer 2 Report

Comments and Suggestions for Authors

Line 10:  prepared to suitable for hot melt processing REMARK:  prepared to be suitable for hot melt processing

Line 14:  Research has found that the E12 content is between 20%wt and 40%wt, the resin system has the best coating viscosity REMARK:  Research has found that when the E12 content is between 20%wt and 40%wt, the resin system has the best coating viscosity

Line 18:  but after more than 50%wt, modulus is almost unchanged REMARK: :  but after more than 50%wt E12 content, modulus is almost unchanged

Line 53: and too little fiber yarn will lead to open cracks REMARK: and too little fiber content will lead to open cracks

Line 58: and is the most widely REMARK and it is the most widely

Line 74: In this study, flexible polyurethane flexible segments were introduced REMARK: In this study, polyurethane flexible segments were introduced

Line 103: According to the stoichiometric number (NCO/OHmolar ratio 2.1:1), the required amount of PPG2000 was weighed REMARK: There was no stoichiometry in this synthesis. Polyurethane reactions are usually not performed with stoichiometric amounts of the raw materials, but using an excess of one of the components, which leads to preferable end groups. It is because the functionality of the raw mterials used is 2 or more.  Please use instead  “calculated amount” Relating to stoichiometry means relating to or denoting quantities of reactants in simple integral ratios, as prescribed by an equation or formula.

Line 114: E44 with stoichiometric number (molar value of 1.1 mol) was added to the prepolymer REMARK: SEE ABOVE

Line 179: while the synthetic product EUR has no absorption peak at this wave number, 179 which indicates that the -NCO group and the epoxy resin side chain. REMARK: NOT UNDERSTANDABLE

Line 181: so that the polyurethane is successfully grafted onto the epoxy resin side chain REMARK: This epoxy resin has no side chains, te side chains are formed by the poluyretane. CORRECT: POLYURETHANE SIDE CHAIN IS SUCCESFULLY GRAFTED ONTO EPOXY RESIN MAIN CHAIN.

Line 221 Figure 4(a) is the tensile stress-strain curve of the resin REMARK: Figure 5(a) is the tensile stress-strain curve of the resin

Comments on the Quality of English Language

Some sentences need to be corrected for clarity. 

Reviewer 3 Report

Comments and Suggestions for Authors

The work provides information on the modification of epoxy resin through additional synthesis with the introduction of polyurethane groups. The mechanical properties of both the new resin and its mixtures with high molecular weight epoxy resin E12 were assessed. The data presented is interesting, but there are a number of questions:

1. A fairly large part of the introduction (and references) is devoted to the practical application of fiber reinforced polymer matrix composites, as well as important aspects of the production and use of prepregs. But unfortunately, there is no data on how new a method is the adding of polyurethane into the resin composition for prepregs (both by mixing and through synthesis).

2. Line 130 “defoamed for 15 min” - Was the mixture foamed or mixed? The composition contains an antifoam agent, but nevertheless in Fig. 7 micrographs show the presence of air bubbles? Why did you foam it? And how much will the presence/absence of such bubbles affect the mechanical properties and crack growth?

3. Line 102 “The flexible polyurethane flexible segment” how relevant is twice flexible?

4. In Fig. 3a shows the IR spectrum of the PU prepolymer for comparison. It is not among the samples indicated in the experimental part. What kind of sample was this? Based on Fig. 1 and the description of the EUR synthesis, it can be assumed that the PU prepolymer is a polyurethane prepolymer, i.e. intermediate in the synthesis of EUR. This point should be clarified in the experimental part (in the description of the synthesis) and the abbreviated name PU prepolymer should be introduced there.

5. Neither in the experimental part nor in the description of the IR spectra is there any indication of how the spectra were processed and comparisons were made. Were the spectra normalized before comparing band intensities (for both the hydroxyl groups and the epoxy group)?

6. In section 3.2 it is written that viscosity was tested in the range of 60-90 C and the disadvantages for prepregs are indicated as the presence of too high or too low viscosity. What is the actual temperature when impregnating prepreg with resin? How will the data obtained relate to actual impregnation? Could you obtain data for the 60%-E12 sample?

7. Line 221 “Figure 4(a) is the tensile stress-strain curve.” Apparently, you meant Figure 5a?

8. Table 2 presents data on mechanical properties. Why does the error in determining Young's modulus increase so much with increasing E12 content?

9. The data in Figure 8 is presented somewhat strangely. If curves for different contents are marked in different colors, then it would be more convenient for analysis and comparison if the color markers were the same in Figures 8a, 8b,c (it would be good if they correlated in color with all the figures for mechanical properties). Why does storage modulus in the range from 20 to 50% and loss modulus for 20 and 30% not have a clear trend with increasing E12 content? Why does Tg determined from Figure 8c fluctuate for different E12 contents?

10. The conclusions should be supplemented with information about what E12 content in rubber systems the authors consider optimal for prepreg impregnation.

Round 2

Reviewer 3 Report

Comments and Suggestions for Authors

Thanks to the authors for the detailed answers to the questions. Some of them were very interesting and it seems to me that some of them could be used to improve the text of the article. For example, the answer regarding viscosity.